# On-Line Monitoring of Pipe Wall Thinning by a High Temperature Ultrasonic Waveguide System at the Flow Accelerated Corrosion Proof Facility

**DOI:** 10.3390/s19081762

**Published:** 2019-04-12

**Authors:** Se-beom Oh, Yong-moo Cheong, Dong-jin Kim, Kyung-mo Kim

**Affiliations:** 1Nuclear Materials Research Division, Korea Atomic Energy Research Institute, 989-111, Daedeok-daero, Yuseong-gu, Daejeon 34057, Korea; ymcheong@kaeri.re.kr (Y.-m.C.); djink@kaeri.re.kr (D.-j.K.); kmkim@kaeri.re.kr (K.-m.K.); 2Department of Materials Science and Engineering, Dankook University, 119, Dandae-ro, Dongnam-gu, Cheonan 31116, Korea

**Keywords:** high temperature pipe, pipe wall thinning, flow accelerated corrosion

## Abstract

Pipe wall thinning and leakage due to flow accelerated corrosion (FAC) are important safety concerns for nuclear power plants. A shear horizontal ultrasonic pitch/catch technique was developed for the accurate monitoring of the pipe wall-thickness. A solid couplant should be used to ensure high quality ultrasonic signals for a long operation time at an elevated temperature. We developed a high temperature ultrasonic thickness monitoring method using a pair of shear horizontal transducers and waveguide strips. A computer program for on-line monitoring of the pipe thickness at high temperature was also developed. Both a conventional buffer rod pulse-echo type and a developed shear horizontal ultrasonic waveguide type for a high temperature thickness monitoring system were successfully installed to test a section of the FAC proof test facility. The overall measurement error was estimated as ±15 μm during a cycle ranging from room temperature to 150 °C. The developed waveguide system was stable for about 3300 h and sensitive to changes in the internal flow velocity. This system can be used for high temperature thickness monitoring in all industries as well as nuclear power plants.

## 1. Introduction

During the operation of nuclear power plants, the thickness of the piping decreases over time which is known as pipe wall thinning. If the reduced thickness is concentrated on one side, the piping could be damaged by the pressure in the pipe, and an internal solution may cause a leak. Pipe wall thinning is mainly caused by FAC (flow accelerated corrosion). FAC occurs mostly in carbon steel pipes, in which the pipe thickness gradually decreases as the Fe ions on the surface of the carbon steel pipe are released. Because pipe wall thinning due to FAC is very slow (a few tens of μm per one year), it is necessary to monitor the piping walls for delamination, cracks and leaks as well as the piping thickness with very high accuracy. This is one of the important issues in the structural stability of a system, which requires continuous monitoring [1,2,3,4,5]. Presently, an ultrasonic method is used, which is one of the nondestructive inspection techniques for measuring the piping wall thickness. The ultrasonic technique is widely used to assess the safety of nuclear piping and to measure the piping wall thickness. Manual ultrasonic methods are generally used to measure the pipe thickness. However, the manual ultrasonic technique has several disadvantages in nuclear power plants. First, the inspection areas of a nuclear power plant are at high temperatures and highly radioactive. Second, the power plant must be shut down and the insulator removed before the thickness of a pipe can be measured because the transducer must be in direct contact with the pipe surface. When the measurement is completed, it is necessary to install the insulation again, so the shutdown time will be longer. Therefore, if the shutdown of the power plant time is prolonged, it will lead to a loss in power production. This process is inconvenient because it is repeatedly measured during a period of time to assess its structural health. Third, the manual ultrasonic thickness measurement method has a low reliability. The accuracy decreases depending on the operator’s skill or condition, the measuring instrument, temperature, the ultrasonic coupling, and the difference in data reading conditions. Therefore, it is necessary to have a stable thickness measurement method for a high temperature and highly radioactive environment without having to stop the operation of the nuclear power plant. 

Measuring the thickness of high-temperature piping using ultrasonic waves has several problems that have not been solved. In the case of a nuclear power plant, the temperature of the fluid flowing inside the pipe increases to about 200 °C. At this time, the piping temperature is up to about 150 °C, and the difference in thermal expansion between each of the piezoelectric and coupling materials may cause errors in the thickness measurements. Conventional piezoelectric materials depolarize if they rise above the Curie temperature; thus, current ultrasonic thickness measurement techniques cannot be used at high temperatures above 200 °C [6,7,8]. 

To solve this problem, a method for installing a buffer rod system and a waveguide method are currently being studied. The buffer-rod system has the advantage of using the existing longitudinal wave transducer; however, it has a disadvantage that it cannot perfectly protect the piezoelectric element from the high temperature piping because the buffer-block is short, and the distance from the specimen is not long. The ultrasonic waveguide system can be used to protect the piezoelectric element from high temperature piping by using a long thin plate to keep the piezoelectric element away from the test specimen [9,10,11,12,13,14]. In this method, the ultrasonic dispersion characteristics in the guide should be considered because the ultrasonic wave propagates through the waveguide [15]. The shear horizontal vibration mode can be used because there is no dispersion characteristic when the wave propagates in the plate. Based on these techniques, we developed a pipe wall thinning monitoring system using a shear horizontal ultrasonic transducer and waveguide strip. Clamping devices were designed and installed with a solid coupling material for safe acoustic contact between the waveguide strip and the pipe surface. The shear horizontal waveguide and clamping device provided an excellent S/N ratio and high measurement accuracy for long time exposure at high temperature conditions.

## 2. Issue of High-Temperature Ultrasonic Thickness Measurements

The ultrasonic thickness measurement principle is usually performed by measuring the flight time between continuous echoes in the time domain. The thickness of the specimen can be determined by calculating the material flight time of the waves and the known ultrasonic velocity values. Assuming that there is minimal ultrasonic dispersion, a sharper ultrasonic signal will increase the resolution of the measurement, and in general is the most accurate way to perform the temporal measurement by measuring the peak to peak time or the perform pulse-echo overlap [16]. Pipe wall thinning of carbon steel pipes in nuclear power plants occurs at several tens of micrometers per year, thus the measurement errors should be minimized. Several factors have been discussed as ways to overcome these errors [17]. First, environmental factors cause errors due to the geometric factors of the piping such as surface roughness, specimen curvature and contact pressure between the coupling material and the transducer as well as ultrasonic velocity errors in the specimen due to changes in the temperature. Thus, the device should be calibrated, and the surface condition should be maintained to minimize measurement errors. Second, there are errors due to the transducer performance and signal processing, such as measurement conditions between the transducer and the specimen, the performance of the analog-to-digital converters and delays caused by digital signal processing, respectively [18]. 

Typical piezoelectric ceramic elements are exposed to temperatures higher than the Curie temperature resulting in depolarization of the element, which then loses its piezoelectric properties making it difficult to accurately measure the signal. The signal quality of the piezoelectric vibrator degrades, and the error in determining the peak position of the signal may increase as the temperature varies. It is necessary to improve the acoustic contact condition between the transducer and the specimen by minimizing the deterioration of the probe at high temperatures. Third, there is a problem with the couplant. For a high temperature pipe, the couplant used at room temperature will evaporate, resulting in an error on the contact surface. Stable ultrasonic sound effect was maintained by using a special high temperature couplant that does not evaporate even in an environment of 150 °C. A dry couplant (gold plate) was used between the waveguide strip and the surface of pipe to minimize the error due to thermal expansion. The gold plate can minimize errors due to thermal expansion between the two objects by constantly maintaining its shape of the coefficient at high temperatures.

## 3. Ultrasonic On-Line Monitoring System for Measuring Wall Thinning of High Temperature Pipe

The conventional buffer-rod type system is widely used as a technique for measuring the thickness of high temperature test specimens. It can be used by inserting a buffer block between the ultrasonic transducer and the specimen shown in Figure 1a. The material of the buffer block should be acoustically stable, should not deform at high temperatures, and should protect the transducer. Glycerin or machine oil used in conventional ultrasonic couplings does not work properly at high temperatures. Therefore, a special solid material coupling, such as a thin gold plate, was used to maintain good acoustic contact between the buffer rod and the specimen. The advantage is that it minimizes the difference in thermal expansion between the buffer rod materials and the specimen, which can be maintained for long periods of the test. Figure 1b shows a buffer-rod type high temperature ultrasonic transducer assembled in a test pipe for thickness monitoring. 

Another approach to measure pipe thickness at high temperatures is to use an ultrasonic waveguide strip. This improved method was attempted using a waveguide strip to reduce the acoustic parameters between the ultrasonic transducer and the specimen. A pair of shear horizontal transducers and a long waveguide strip were designed and manufactured. The shear horizontal vibration mode was chosen to ensure that there was proper ultrasonic wave transmission at the thin strips. The shear horizontal mode had sharp and clear ultrasonic signals within a certain frequency range because there was low dispersion in the plate. This vibration mode is advantageous for obtaining sensitive and accurate experimental data at high temperatures [19]. The shear horizontal wave transducer was attached to the edge of the waveguide strip shown in Figure 2. When the transducer and the waveguide strip contacted each other exactly at a perpendicular level, the shear horizontal mode was stably transmitted to the waveguide. On the opposite side of the waveguide strip, a clamping device was designed and fabricated to precisely hold the specimen, and two waveguide strips were installed in parallel to divide the transmitter and receiver. A thin solid plate (gold plate) was used as a couplant between the waveguide strip and the test specimen similar to the buffer-rod system. The transducer used a waveguide strip that was far from the specimen, which was maintained at about 35 °C when the temperature of the pipe was 150 °C. This meant that the developed system completely freed the transducer from the constraints of high temperatures. The waveguide pitch/catch method using two waveguide strips can increase the S/N ratio compared to the pulse/echo technique. Two strips are used to set up the waveguide system device. A transducer was connected to each strip, one on the transmitting transducer and the other on the receiving transducer. The two strips were spaced 1 mm apart to filter the noise received at the surface of the pipe to be measured. If the strip is placed at 1 mm, it measured only the wanted signal because of the time difference between the signal transmitted to the pipe surface, and the signal reflected from the opposite side of the pipe. Because it received a signal only from the piping specimen, noise from undesired reflections at the end of the strip were prevented. This method had no main bang signal, and the signal reflected from the end of the waveguide strip was very small. Additionally, the multiple reflected signals on the back wall of the pipe, which was to be inspected, had a high S/N ratio. 

## 4. High Temperature Pipe Thickness Measuring Program

The thickness measurement program was designed as a moving gate in real time to accurately measure the reflected flight time. The first gate was set to the signal from the end of the transmitting waveguide strip shown in Figure 3. The second gate was set to the first back wall echo signal, and the third gate was set to the second back wall echo signal. The second and third gates were set as the moving gates to follow the first gate setting. The moving gate moves along with the movement of the rf signal according to the noise or the temperature change, making it possible to measure the peak to peak signal stably. The peak of the first and second back wall echoes were automatically determined by the flight time and denoted as t_1_ and t_2_ shown in Figure 3. The ultrasonic wave velocity was constant and the path length of the actual reciprocating wave can be seen by the strip and pipe wall thickness. Thus, the time of the received rf signal can be calculated. The shear horizontal wave velocity of the carbon steel is about 3300 m/s, and the flight time reflected by the waveguide strip 300 mm in length is calculated to be 170 μs. The flight time between the first back wall and the second back wall of a 5.54 mm thick pipe is estimated to be about 3.2 μs. All ultrasonic rf (radio frequency) waveforms are in the time domain and displayed on the PC screen. Moreover, this system inspects the signal quality and is designed to display an alarm indicator on the screen when receiving unwanted signals. Between t_1_ and t_2_, the flight time was automatically calculated on average, hundreds of times to obtain accurate thickness data. Because ultrasonic velocity is a function of temperature, variation in the ultrasonic velocity at high temperatures can be a main problem in terms of measurement data errors. Therefore, to measure the thickness in real time at high temperature, pre-calibration is required to reflect the relationship between the ultrasonic velocity and the temperature.

The ultrasonic velocity can be determined as a function of the temperature by measuring the variation in the velocity with a temperature change in the material to be measured. Figure 4 shows the measurement data of the shear mode wave velocity in a carbon steel pipe based on a variation in the temperature. The velocity of the ultrasonic waves was measured by heating the same materials pipe as the pipe used in the FAC proof facility to the furnace from 0 to 250 °C. The pipe thickness did not change when measuring the sound velocity change by calculating the time of the received signal according to the temperature change [20]. This function was entered into the thickness measurement program to reduce the error due to the temperature variation.

## 5. Verification Experiment in the FAC Proof Facility and Results

The FAC demonstration test facility was manufactured to operate in the same environment as a nuclear power plant. This facility is designed to operate for 1200 h at high temperatures at more than 150 °C, per one cycle and with the adjustable pH, DO, and flow rate for the fluid flowing inside the piping. To measure pipe wall thinning, we prepared a test section with the insulation removed from part of the facility. This section is made of carbon steel (SA 106), and the pipe has an outer diameter of 60.4 mm, a wall thickness of 5.54 mm and a length of 750 mm. The chemical composition of the materials is shown in Table 1. The pipe thickness was measured by the buffer-rod type system and an ultrasonic waveguide high temperature thickness monitoring system. The two systems were installed on the surface of the carbon steel pipe to compare the signals from each other, as shown in Figure 5. Prior to the experiment, the thickness measurement error was confirmed according to the temperature variation for the stability and accuracy of the system. Figure 6 shows the actual measured waveguide system data. A provisional thickness error range was determined by increasing the temperature of the same material pipe to 200 °C and measuring the change in thickness during the cooling process. The range of the thickness measurement error was ±15 μm while the pipe was heated to 0~150 °C and then cooled. These devices were able to acquire ultrasonic signals which were reliable for a long time at high temperatures, and the flight times of the signals through calculations were converted to the thickness of the pipe. Minimizing the measurement errors was possible with normalization of the signal amplitude, the automatic setting of the ultrasonic flight time and moving the gate control by applying a temperature compensation factor. The pipe thickness data were directly compared with the measured data at room temperature during the rest period to determine the reliability of the measured data. 

Figure 7 shows the change in the slope of the wall thickness reduction with the flow rate in the waveguide system. This developed system measured the tendency of pipe wall thinning by changing the flow rates from 7 to 10 to 12 m/s every 1100 h. The blue line is the data measured in real time once every hour. The red line slope indicated the FAC rate in each section. As the flow rate increased, the FAC rate became faster and the FAC rate decreased as the flow rate decreased. As the flow rate increased from 7 m/s to 12 m/s, the wall thickness reduction rate also increased. This result meant that is possible to predict the change in the flow rate inside the pipe by analyzing the rate of pipe wall thinning.

Figure 8 shows the pipe wall thinning measurement data from approximately 3300 h of operation using the buffer-rod and developed waveguide systems. The temperature of the fluent flowing inside the pipe was maintained at 150 °C, and both devices measured the wall thickness reduction of about 260 μm. The blue line in Figure 8 is the data of thickness thinning of buffer-rod type commercial equipment. From the point at which the flow rate increased from 10 m/s to 12 m/s, it can be seen that the thickness thinning rate tendency was less than that of the developed equipment. Furthermore, although the flow velocity slowed to 7 m/s at a large velocity of 12 m/s, but the buffer-rod type system was measured to show that the thickness reduction rate was increasing. On the other hand, the red line is measuring data by the developed waveguide system. The thickness reduction rate was measured to suit the change in the flow rate. It is possible to compare more clearly converted (mm/year) for the thickness reduction for each period. As the flow rate inside the piping increased, the rate of the pipe wall thinning increased. The developed waveguide system showed more accurate reduction trends as the flow rate changed. The developed waveguide system operated stably at a temperature cycle of 150 °C for a long time, and the measurement error was about ±20 μm. 

Figure 9 shows the result of the manual UT measurement at room temperature. A total of six points were set in the same direction as the position where the two systems were installed, and measurements were made at room temperature. Reliable comparative data measures exactly at the same point, but there was an error generated by removing and reinstalling real-time measuring equipment. Measuring the six zones from A to F next to the point where two pieces of equipment were installed (see Figure 10). A total of six points were measured 10 times and averaged in order to increase the accuracy. The equipment used for the measurement was 38DL PLUS (OLYMPUS). The tendency of the thickness reduction measured with the manual room temperature UT was very similar to that for the high temperature on-line UT system. The reliability of the developed waveguide system was verified by comparing the conventional buffer-rod system and the manual room temperature UT.

## 6. Conclusions

A shear horizontal ultrasonic pitch/catch waveguide system was developed for the accurate online monitoring of the pipe wall thickness in the FAC certification test facility. A clamping device was designed and installed for the gold-plate contact between the end of the waveguide strip and the pipe surface. A computer program was developed for online monitoring of the pipe thickness at high temperatures. The system minimized measurement errors by controlling the moving gate with temperature deviation, normalizing the signal amplitude, automatically determining the ultrasonic flight time and including a temperature compensation function. The buffer-rod and ultrasonic waveguide high temperature ultrasonic thickness monitoring systems were successfully installed in the test section of the FAC test facility. These systems were confirmed as a stable operation with an error of ±20 μm in temperature cycles up to 150 °C for 3300 h, and performed better than other similar measurement systems. In addition, the developed waveguide system was able to predict the velocity of the fluid flowing inside the pipe by analyzing the thickness reduction rate. 

Finally, it was confirmed that the thickness reduction measurement was very accurate in comparison with the room temperature UT results. This result demonstrates that a waveguide system is sensitive to the flow of internal fluids and can measure thickness better than that of commercial systems. This system can be applied to high temperature thickness monitoring in all industries as well as nuclear power plants.

## Figures and Tables

**Figure 1 sensors-19-01762-f001:**
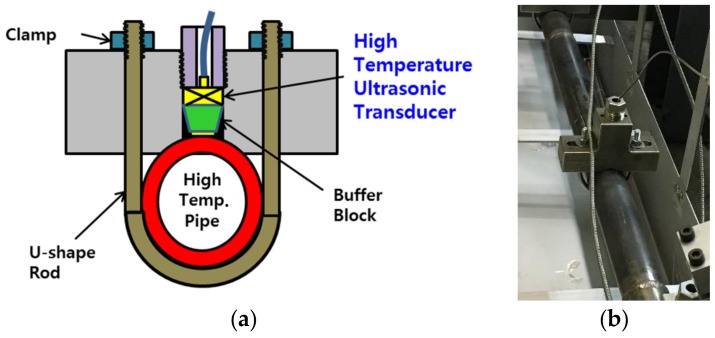
(**a**) An assembly drawing diagram of a high temperature pipe using a buffer-rod type measurement system; (**b**) an installed buffer-rod type system for thickness monitoring on a pipe.

**Figure 2 sensors-19-01762-f002:**
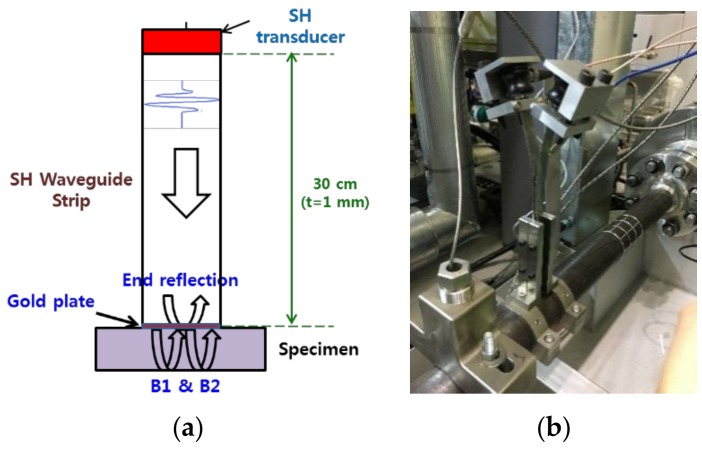
(**a**) Conceptual diagram of a pair of waveguide strips for high temperature thickness monitoring; (**b**) installed waveguide system for thickness monitoring on a pipe.

**Figure 3 sensors-19-01762-f003:**
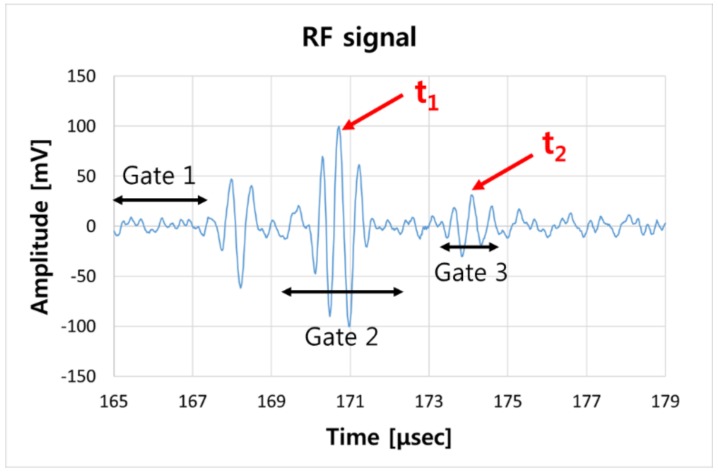
Typical ultrasonic rf signals by a developed waveguide with the pitch/catch method at 150 °C.

**Figure 4 sensors-19-01762-f004:**
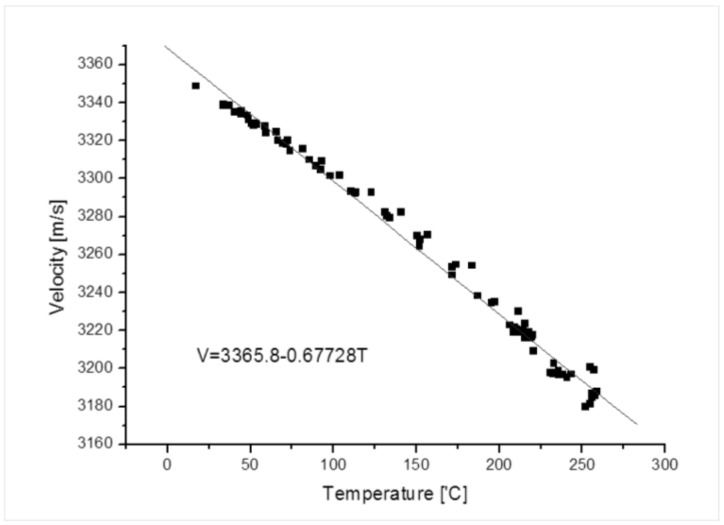
Calibration of the shear horizontal wave velocity with varying temperatures for a carbon steel pipe SA 106.

**Figure 5 sensors-19-01762-f005:**
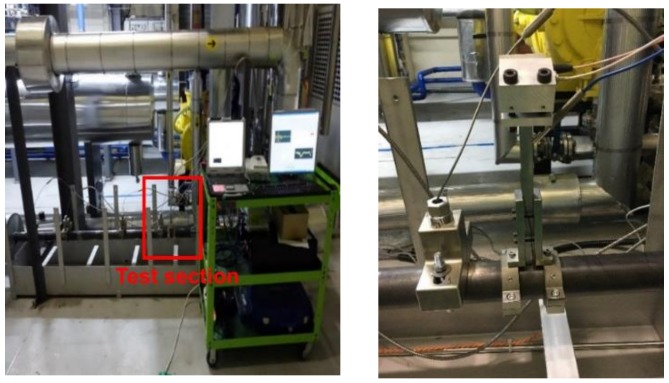
Test section in the flow accelerated corrosion (FAC) facility (**left**) and another type system installed at a test section (**right**).

**Figure 6 sensors-19-01762-f006:**
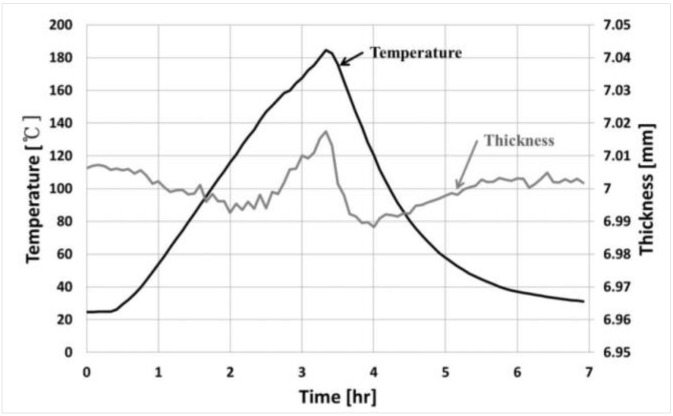
The measurement error by temperature variation in the developed system.

**Figure 7 sensors-19-01762-f007:**
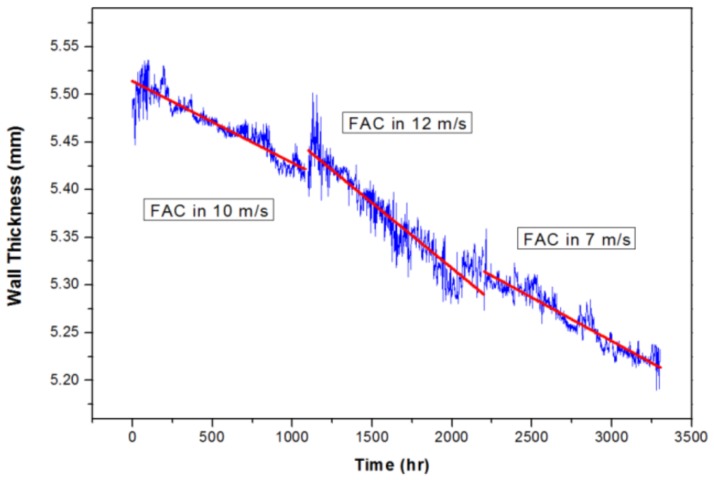
Pipe wall-thickness monitoring of carbon steel piping in the flow accelerated corrosion proof test facility: Different wall-thinning ratios observed depending on the flow velocities.

**Figure 8 sensors-19-01762-f008:**
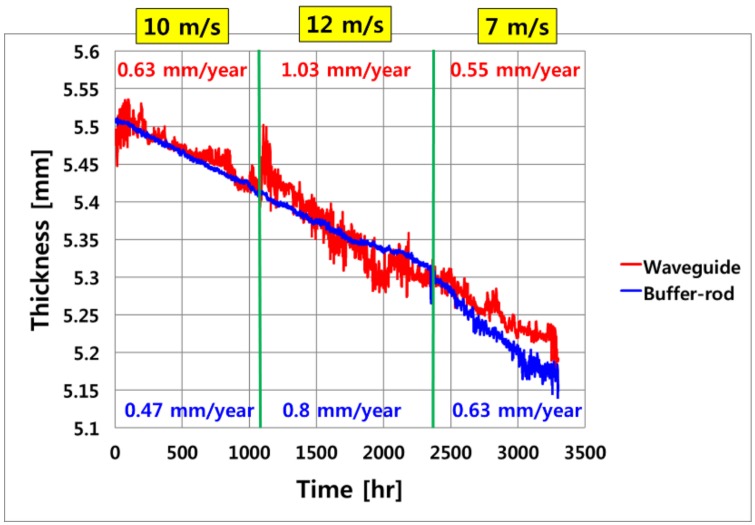
The pipe thickness measurement data using the buffer-rod and waveguide type systems.

**Figure 9 sensors-19-01762-f009:**
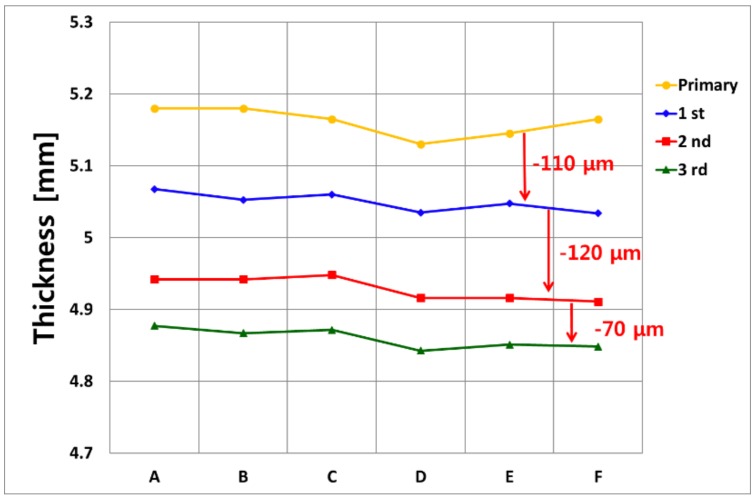
The thickness measurement result of the manual ultrasonic testing (UT) at room temperature.

**Figure 10 sensors-19-01762-f010:**
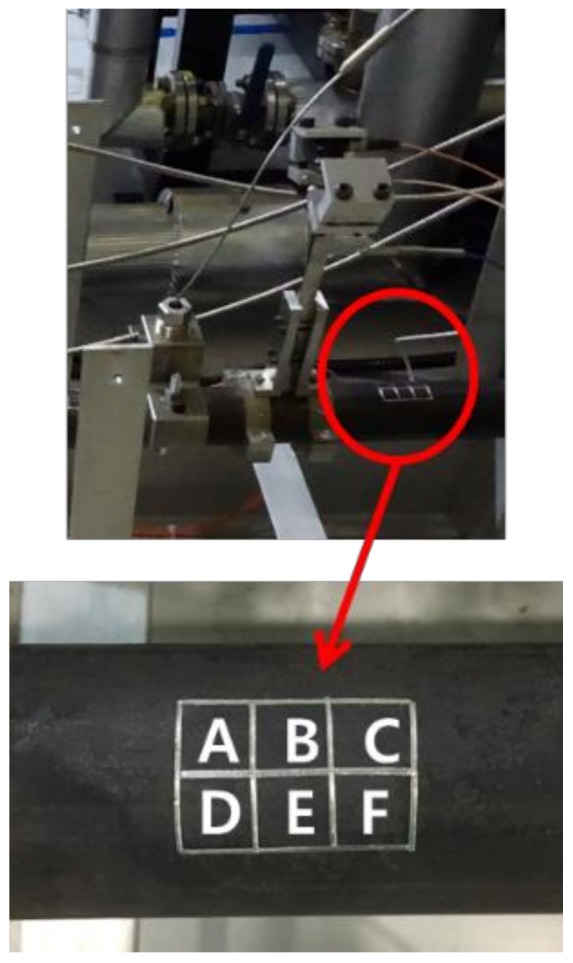
A grid for measuring ultrasonic testing at room temperature in the same direction next to two installed systems.

**Table 1 sensors-19-01762-t001:** The chemical composition of the material (SA 106 Gr. B).

Material	Cr	Mo	Cu	Mn	Ni	Si	C	P	S
SA106 Gr. B	0.02	0.01	0.04	0.37	0.02	0.22	0.19	0.008	0.006

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
