# Peer review of "On-Line Monitoring of Pipe Wall Thinning by a High Temperature Ultrasonic Waveguide System at the Flow Accelerated Corrosion Proof Facility"

_sensors, 2019, doi:10.3390/s19081762_

Round 1

Reviewer 1 Report

The paper proposes a shear horizontal ultrasonic pitch/catch technique for inspection of pipe wall-thickness at in high temperature conditions. It is suitable for nuclear power plants. The work is well organized and developed. The results support the conclusions appropriately. It is a neat experimental work and the results are satisfactory enough. The work should be accepted. Two small revisions:

1)      How do you calculate the time of flight? This seems to be challenging from Figure 3. Nowadays, there are advanced techniques for this purpose, based on advanced signal processing techniques like wavelet.

2)      Thanks to include this paper into your references:

C. Abarkane, D. Galé-Lamuela, A. Benavent-Climent, E. Suárez, A. Gallego.

Ultrasonic pulse-echo signal analysis for damage evaluation of metallic slit-plate hysteretic dampers. Metals (MDPI), 2017, DOI: 10.3390/met7120526

Author Response

Thank you for your interest in this manuscript and your good comments and answers.

1)      How do you calculate the time of flight? This seems to be challenging from Figure 3. Nowadays, there are advanced techniques for this purpose, based on advanced signal processing techniques like wavelet.

->It is possible to calculate the time of flight through the ultrasonic wave velocity by the total length passing through the strip and pipe wall. More details have been added to the manuscript.

I have explained more detail in the manuscript. I hope this is the answer to your question.

Thank you

Yours sincerely

Reviewer 2 Report

The article presents an interesting design of the buffer-rod type system for thickness monitoring of high-temperature pipes. The proposed solution presents high application values, and verification tests confirmed its effectiveness.

1) "Conventional piezoelectric materials depolarize if they rise above the Curie temperature; thus, current ultrasonic thickness measurement techniques cannot be used at high temperatures above 200 ˚C"

Comment: High-temperature piezoelectric AE sensors are available on the market. An example is the sensor PAC D9215, which operates in the temperature range from -200°C to 540°C. This type of transducer has been adapted for use in high temperature, radiation environments such as in nuclear power plants and for monitoring high-temperature equipment in power plants, pipelines. What is the advantage of the proposed solution compared to the high-temperature AE sensors available on the market.

2) "All ultrasonic rf (radio frequency) waveforms are in the time domain and displayed [...]"

Comment: The term "radio frequency RF" refers to electromagnetic waves. It should not be used to describe sound waves (including ultrasonic waves).

3) The article does not contain keywords.

4) "The chemical compositions of the materials are shown in Table 1."

Comment: There is no table in the article.

Author Response

Thank you for your interest in this manuscript and your good comments and answers.

1) "Conventional piezoelectric materials depolarize if they rise above the Curie temperature; thus, current ultrasonic thickness measurement techniques cannot be used at high temperatures above 200 ˚C

-> In the case of AE sensors, it is a good idea to evaluate the material integrity such as nuclear power plants. The AE sensor is very sensitive to changes in sound when a signal is generated by bubbles, cracks, or corrosion products on the pipe surface. However, it is difficult to determine the precise location of the occurrence, and it is difficult to apply to structural variation such as thickness thinning. This study focuses on the development of new sensor than AE sensor because it is to monitor pipe thickness thinning in real time. In addition, if the sensor is exposed to a high temperature pipe for a long time, the life of the sensor can be shortened.

2) "All ultrasonic rf (radio frequency) waveforms are in the time domain and displayed [...]"

Comment: The term "radio frequency RF" refers to electromagnetic waves. It should not be used to describe sound waves (including ultrasonic waves).

->The rf signal used in this paper is the term used in ultrasonic testing to represent the waveform of a signal received in the time domain.

Other comments have been modified by adding content to the manuscript.

I hope this is the answer to your question.

Thank you

Yours sincerely

Reviewer 3 Report

The authors presented an innovative system to measure the thickness of a pipe based on a piezoelectric sensor which is able to work even at high temperature. The approach adopted to avoid overcoming the Curie temperature at the sensor is based on a special strip which is a waveguide for the emitted ultrasound. The pulse- echo approach or the pitch catch approach while using two strips is adopted to quantify the effective thickness.

Although the results are quite promising, the authors are encouraged to address a few major comments listed above.

1.       Some mistakes including typos, grammatical errors and complex sentences are present in the manuscript. For example, is decreases on line 45, sentence on line 107-109, Table 1 is missing, etc. Please proofread the article.

2.       The authors state that “An ultrasonic thickness measurement method at high temperatures has not been developed yet.” However there is a wide literature dealing with this issue. The authors are encouraged to revise the state of the art presented in a discussing-mode with the attempt to realize the novelty of the system (rather than method) presented.

3.       The authors state that: “Generally, the most accurate method for time measurements is to measure peak-to-peak times or the pulse echo overlap [16].” This is valid only when no or low dispersivity is present in the excited mode. Otherwise this approach lead to evaluate phase velocity. Please, explain this aspect while discussing the reference.

4.       The authors discussed several sources of error that should be minimized to have a reliable thickness quantification. In addition to those, another source of error is described in: Maio et al., Application of laser Doppler vibrometry for ultrasonic velocity assessment in a composite panel with defect (2018) Composite Structures, 184, pp. 1030-1039. It is worth discussing this aspect and citing the reference.

5.       Sentence on lines 93-97 is not clear. Please, discuss more in detail this aspect which represents a crucial part of the system.

6.       On line 127, the double strip option is introduced. Please, discuss in detail its setup and different way to deal with signals.

7.       From line 141 a certain number of data are presented, including the SH wave velocity if the carbon steel. However, it is not clear how it is derived or calculated there. Likewise, it is not clear how all other data reported are estimated. Please, specify if the method needs a calibration, the definition of a material or something else before using it to define the specimen thickness. All those aspects are confusing.

8.       The authors state that the acquired ultrasonic signals were modified.  How do they modify them? Please specify.

9.       Plot in Figure 4 is measured on the specimen under investigation or taken from ref 19? Please add details about that.

10.   Which is the predicted slope for each of the three ranges shown in Fig. 7? It is worth comparing those with the FAC values.

11.   The authors state that: “The developed waveguide system showed more accurate reduction trends as the flow rate changed.” Please, better discuss and motivate this consideration.

12.   In fig 9 it is not clear where the points are located? It is worth comparing the results obtained manually with those obtained with the proposed system in the same points. Such comparison should appear together with a detailed discussion.

13.   The authors conclude stating that: “The system minimized measurement errors by controlling the moving gate with 220 temperature deviation, normalizing the signal amplitude, automatically determining the ultrasonic 221 flight time and including a temperature compensation function.” However this aspect is not stressed enough in the manuscript.

14.   It is not clear how the error of 20 microns is estimated.

Author Response

Thank you for your interest in this manuscript and your good comments and answers. I expect that will be able to complete my manuscript through your good comments and reviews. Thank you for recommending a good reference paper. It was a good reference for completing the manuscript. All comments have been modified by adding content to the manuscript.

I hope this is the answer to your question.

Thank you

Yours sincerely,

Round 2

Reviewer 3 Report

The authors improved  the mauscript and the revised paper is acceptable for publication. 

Sensors EISSN 1424-8220 Published by MDPI AG, Basel, Switzerland RSS E-Mail Table of Contents Alert
Back to Top